# Hemp Seed Oil in Association with β-Caryophyllene, Myrcene and Ginger Extract as a Nutraceutical Integration in Knee Osteoarthritis: A Double-Blind Prospective Case-Control Study

**DOI:** 10.3390/medicina59020191

**Published:** 2023-01-18

**Authors:** Giacomo Farì, Marisa Megna, Salvatore Scacco, Maurizio Ranieri, Maria Vittoria Raele, Enrica Chiaia Noya, Dario Macchiarola, Francesco Paolo Bianchi, Davide Carati, Simona Panico, Eleonora Di Campi, Antonio Gnoni, Venera Scacco, Alessio Danilo Inchingolo, Erda Qorri, Antonio Scarano, Biagio Rapone

**Affiliations:** 1Department of Translational Biomedicine and Neuroscience (DiBraiN), Aldo Moro University, 70121 Bari, Italy; 2Department of Biological and Environmental Science and Technologies (Di.S.Te.B.A.), University of Salento, 73100 Lecce, Italy; 3Mater Dei Hospital C.B.H., 70125 Bari, Italy; 4Istituti Clinici Scientifici Maugeri—IRCCS Bari, 70124 Bari, Italy; 5Department of Biomedical Science and Human Oncology, Aldo Moro University of Bari, 70121 Bari, Italy; 6Ansce Bio Generic, 73020 Carpignano Salentinono, Italy; 7Interdisciplinary Department of Medicine, University of Bari “Aldo Moro”, 70121 Bari, Italy; 8Dean Faculty of Medical Sciences, Albanian University, 1001 Tirana, Albania; 9Department of Oral Science, Nano and Biotechnology, CaSt-Met University of Chieti-Pescara, 66100 Chieti, Italy

**Keywords:** knee osteoarthritis, rehabilitation, dietary supplement, hemp seed oil, terpenes

## Abstract

*Background and Objectives:* Nutraceuticals are gaining more and more importance as a knee osteoarthritis (KOA) complementary treatment. Among nutraceuticals, hemp seed oil and terpenes are proving to be very useful as therapeutic support for many chronic diseases, but there are still few studies regarding their effectiveness for treating KOA, both in combination and separately. The aim of this study is thus to compare the effect of two dietary supplements, both containing hemp seed oil, but of which only one also contains terpenes, in relieving pain and improving joint function in patients suffering from KOA. *Materials and Methods:* Thirty-eight patients were recruited and divided into two groups. The control group underwent a 45 day treatment with a hemp seed oil-based dietary supplement, while the treatment group assumed a hemp seed oil and terpenes dietary supplement for the same period. Patients were evaluated at the enrollment (T0) and at the end of treatment (T1). Outcome measures were: Numeric Rating Scale (NRS), Oswestry Disability Index (ODI), Short-Form-12 (SF-12), Knee Injury Osteoarthritis Outcome Score (KOOS), and Oxford Knee Score (OKS). *Results:* All outcome measures improved at T1 in both groups, but NRS, KOOS and OKS had a greater significant improvement in the treatment group only. *Conclusions:* Hemp seed oil and terpenes resulted a more effective integrative treatment option in KOA, improving joint pain and function and representing a good complementary option for patients suffering from osteoarthritis.

## 1. Introduction

Knee osteoarthritis (KOA) is the most common form of limb osteoarthritis [1]. It is a chronic joint disease which is characterized by degenerative lesions of the articular cartilage that progressively cause pain, motor impairment and, in the most severe cases, deformation of the joint itself [2,3]. Although one of the main risk factors is old age, as KOA predominantly affects people in their 70s [4], KOA is also increasing in patients aged between 45 and 70 [5], emerging recently as a very relevant problem for society and placing it among the most common causes of disability [6]. At present, KOA remains an untreatable condition because its mechanisms of progression are not fully understood [7].

Therefore, the goal of osteoarthritis treatment is to alleviate symptoms and to slow down the disease progression. The KOA therapeutic spectrum ranges from pharmacotherapy to physical therapies, orthotics and, finally, surgery and rehabilitation [8,9,10,11,12]. In recent years, nutraceuticals, which are dietary supplements used to improve health, delay aging, prevent disease, and support the functioning of the human body, are gaining importance [13]. In patients with osteoarthritis, the assumption of long-chain omega-3 fatty-acids from fish oil supplements and micronutrients such as vitamin K is considered very useful [14], since it has a role in bone and cartilage mineralization. New molecules are constantly emerging, including components of industrial hemp (Cannabis sativa, Cannabacae), which have already been shown to be effective for anxiety disorders and in reducing oxidative stress, contrasting the risk of chronic diseases including joint diseases, neurological disorders, digestive problems and skin conditions [15,16]. Recent studies showed important preclinical and clinical evidence about Cannabis Sativa pain relief properties [17,18], especially because it contains two main phytocannabinoids: D9-tetrahydrocannabinol (D9-THC) and cannabidiol (CBD) [19,20].

Among the several components of Cannabis Sativa, as flavonoids, vitamins, fatty acids, sterols, lignanamides, spiroindans, and alkaloids that may have health benefits [21], terpenes represent a very attractive option for pain treatment [22]. In particular, β-caryophyllene (BCP), a bicyclic sesquiterpene very present in Cannabis Sativa, has been widely investigated and highly appreciated for its low toxicity and considerable safety profile [23]. One of its main targets was described to be the cannabinoid receptor type 2 (CB2 receptor), for which it is thought to act as a full agonist [24]. Interestingly, recent data suggest that the selective agonism of CB2 receptors may constitute a novel strategy for treating chronic pain [25]. β-Myrcene is a monoterpene composed of two isoprene units, and a recent study showed its significant anti-inflammatory and anti-catabolic effects in human chondrocytes and, thus, its ability to halt or, at least, slow down cartilage destruction and osteoarthritis progression [26]. Ginger extract takes advantage of the anti-inflammatory properties of gingerols and shagaols, which selectively inhibit COX-2 (cyclooxygenase-2) and inflammatory cytokines [27]. Nevertheless, there remains little evidence with regard to the effects of Cannabis Sativa components on osteoarthritis pain management, especially from a clinical point of view.

Thus, the aim of this study is to compare the effect of two regimens of food supplementation in relieving pain and to improve joint function in patients aged between 45 and 70 suffering from KOA: the first one based exclusively on hemp seed oil (without cannabinoids), the second one based on hemp seed oil (without cannabinoids) but potentiated with terpenes (β-caryophyllene and myrcene). This comparison could be useful to understand if the two nutraceuticals are useful and if one of these is better than the other, considering their different composition.

## 2. Materials and Methods

The study model is a double-blind prospective case-control study. The study was carried out in the period between March and August 2022. Patients were enrolled if they met the following criteria: age between 45 and 70 years; a clinical diagnosis of KOA according to the American College of Rheumatology criteria; knee pain ≥ 4 according to the Numeric Rating Scale (NRS) at the enrollment and in the previous 15 days; radiographic KOA classifiable as grade II-III according to Kellgren-Lawrence scale; ability to understand the purpose and design of the study, and to provide informed consent. Exclusion criteria were: KOA local complications (e.g., hematoma and joint effusion); knee pain due to trauma (during the previous month); any disease potentially interfering with medical evaluation different from KOA (e.g., rheumatoid arthritis, metabolic inflammatory arthropathy); local drug infiltration (hyaluronic acid, steroids, stem cells, polynucleotides, Platelet Rich Plasma) or physiotherapy (e.g., laser therapy, shock wave therapy, therapeutic exercise, etc.) within the previous 45 days; assumption of non-steroidal anti-inflammatory drugs or analgesics within 15 days prior the enrollment; assumption of slow-acting drugs or dietary supplements in the previous 3 months (e.g., chondroitin sulfate, diacerein, soybean and avocado unsaponifiables, oxaceprol, granions de cuivre, glucosamine, phytotherapy for osteoarthritis); contraindications to acetaminophen; systemic diseases which contraindicate nutraceuticals assumption (liver failure, kidney failure, uncontrolled cardiovascular disease); pregnant or lactating women; pre-menopausal women not using contraception; and patients enrolled in other clinical trials within the past three months.

Thirty-eight patients with monolater KOA were recruited and then divided into two groups, each consisting of nineteen subjects.

At the time of recruitment (T0), all patients underwent a medical examination, which included medical history, standardized physical examination, and x-rays evaluation. Therefore, the weight and height of each patient were detected and the Body Mass Index (BMI) was calculated according to the formula: weight (Kg)/height (m^2^). The following rating scales were then measure for each patient:Numeric Rating Scale (NRS): this is a one-dimensional scale that rates pain from 0, the absence of pain, to 10, the maximum perceived pain;Oswestry Disability Index (ODI): this is a scale that rates the percentage value of disability and ranges from 0%, no disability, to 100%, maximum disability;Short Form 12 (SF-12): this is a quality of life assessment scale. It is divided into physical domain (PCS) and mental domain (MCS). The higher the score, the better the patients’ quality of life;Knee Injury and Osteoarthritis Outcome Score (KOOS): this is a percentage value that quantifies clinical symptoms, disability, and quality of life in patients suffering from knee diseases. It ranges from 0% (severe disability) to 100% (optimal condition);Oxford Knee Score (OKS): this assesses the severity of osteoarthritis from 0 to 48 (severe osteoarthritis 0–19; moderate-severe osteoarthritis 20–29; mild-moderate osteoarthritis 30–39; no sign of osteoarthritis 40–48).

Patients belonging to the control group underwent a 45 day treatment with a dietary supplement based on hemp seed oil (413 mg/capsule) in a softgel capsule format.

Patients belonging to the treatment group underwent a 45 day treatment with a dietary supplement based on hemp seed oil (413 mg/capsule), β-caryophyllene (35 mg/capsule), myrcene (15 mg/capsule), and ginger extract titrated in gingerols (66 mg/capsule). The hemp seed oil contained in both dietary supplements was composed mainly of Linoleic (55.90%), gamma-Linolenic (19.10%) and Oleic (9.30%) acids. Patients were unaware of which of the two dietary supplements they were taking as they were not identifiable from the packaging. Similarly, the physicians performing the clinical assessments at T0 and T1 were unaware of which supplement the patients had taken, thus creating a double-blind study design. A third investigator was therefore responsible for the distribution of the supplements.

All patients took two softgels of the assigned dietary supplement per day, one capsule with each main meal (usually during lunch and dinner).

Patients were allowed to take paracetamol (up to a maximum of 3000 mg/day) and were asked to write down the dosage of the drug taken in a dedicated diary. The use of other medications during the treatment period was recorded, as were the eventual side effects.

At the end of the treatment (T1), 45 days after T0, all of the outcome measures were collected from each patient in order to compare the clinical trend and the functional implications between the groups. The diaries used to register any drugs taken in addition to any side effects related to the proposed therapies were collected at the same time.

All patients received the necessary information during the first medical examination and expressed their written informed consent. All of the performed procedures were carried out in accordance with the Helsinki Declaration (2016) of the World Medical Association. The study was approved by the Ethics Committee of Albania University, Tiran, Albania (Nr. 587 Prot.–Date: 13 December 2021).

### Statistical Analysis

A data analysis was performed using STATA MP17 software. Continuous variables were described as mean ± standard deviation (SD) and range, and categorical variables as proportions. A skewness and kurtosis test was used to evaluate the normality of continuous variables and a normalization model was constructed using the logarithmic function for those not normally distributed. The Student’s *t*-test for independent data was used to compare continuous variables between groups, and the ANOVA for repeated measures test was used to compare continuous variables between groups and detection time. Multivariate linear regression was used to assess the relationship between the difference from T1 to T0 of each individual outcome and the group (treatment vs. control), sex (male vs. female), age (years) and BMI; correlation coefficients were calculated, with a 95% confidence interval (95%CI) indicated. A *p*-value < 0.05 was considered significant for all tests.

## 3. Results

The study sample was made up of 38 subjects, of which 19 (50.0%) belonged to the control group and 19 (50.0%) belonged to the treatment group; the characteristics of the sample, by group, are shown in Table 1.

The outcome variables, by group and detection time, are described in Table 2 and Figure 1, Figure 2, Figure 3, Figure 4, Figure 5 and Figure 6; the ANOVA test for repeated measures showed a statistically significant difference for all the outcome measures in the comparison between T0 and T1 (*p* < 0.0001). The same test showed a statistically significant difference for NRS, KOOS and OKS scores in the interaction between T0 and T1 and between the two groups (*p* < 0.0001). All of these findings are described in Table 2.

Table 3, Table 4, Table 5, Table 6, Table 7 and Table 8 describe the multivariate linear regression analyses by single outcome. Specifically, in Table 3 a statistically significant improvement in the NRS scores emerged between T0 and T1, attributable solely to the treatment (*p* < 0.0001).

In Table 4, a statistically significant improvement in the ODI scores emerged between T0 and T1, attributable exclusively to patients’ BMI (*p* = 0.036).

In Table 5, a statistically significant improvement in the PCS-12 scores emerged between T0 and T1, attributable to the treatment (*p* < 0.036) and to BMI (*p* < 0.036).

In Table 6, no statistically significant differences were found between T0 and T1 for MCS-12 values. However, it should be noted that the starting values were already uneven between the two groups, especially with regard to the MCS-12. This is attributable to the fact that this rating scale, especially for the mental dimension, is easily influenced by factors other than knee pain.

In Table 7, a statistically significant improvement in the KOOS scores emerged between T0 and T1, attributable exclusively to the treatment (*p* < 0.0001).

In Table 8, a statistically significant improvement in the OKS scores emerged between T0 and T1, attributable exclusively to the treatment (*p* < 0.0001).

From the analysis of analgesic intake diaries, only a random intake emerged, which settled on an average of 1.0 g/week per group, with a sporadic and not significant distribution among the participants. No side effects were referred.

## 4. Discussion

The aforementioned results showed that both dietary supplements produced beneficial effects in patients. However, NRS, KOOS and OKS scores had a statistically significant greater improvement in the treatment group, which is the one treated with the dietary supplement containing both hemp seed oil and terpenes. Therefore, this latter seems more effective in relieving KOA pain and improving specific knee function.

As was said previously, hemp seed oil accounts for the increasing scores in both groups [15,16], but in assessing the composition of the two dietary supplements it is likely that the better results of the treatment group, both in terms of pain relief and in terms of joint function, derive from the presence in the one taken by this group of terpenes, more specifically BCP and myrcene. In fact, in 2020, Rao Jiang-Yan et al. showed how, through autophagic activation, BCP is able to alleviate cerebral ischemia/reperfusion injury in mice, highlighting its protective role in animal cells and vessels [28]. Experimental studies showed that BCP reduces pro-inflammatory mediators such as tumor necrosis factor-alfa (TNF-α), interleukin-1β (IL-1β), interleukin-6 (IL-6), and nuclear factor kappa-light-chain-enhancer of activated B cells (NF-κB), thus ameliorating chronic pathologies characterized by inflammation and oxidative stress [29,30,31,32]. In 2012, Ou Ming-Chiu et al. recruited 48 women with primary dysmenorrhea with an NRS > 5. On menstrual cycle days, twenty-four patients massaged synthetic fragrances on the abdomen and the other twenty-four patients did the same with BCP-based essential oils. Abdomen massages with BCP-based essential oils provided relief to patients with primary dysmenorrhea, and the duration of menstrual pain was also reduced [33]. Moreover, Shim Ik Hyun et al. showed the effectiveness of BCP in reducing Helicobacter Pylori related gastritis, particularly nausea and epigastric pain [34]. Due to its lipophilicity, BCP is highly lipophilic, so it possesses a good oral bioavailability [35]. Ibrahim et al. demonstrated BCP’s ability to act as a significant antinociceptive without any damage to gastric mucosa [36]. In addition, BCP is able to reduce the expression of COX-2 and inducible Nitric Oxide Synthase (iNOS), avoiding NF-κB activation, so analgesia is consequently achieved [37]. The reduction of acute and chronic pain is achieved by BCP because of its interaction with the opioid system [38]. In fact, BCP promotes the release of β-endorphin secondarily affecting the opioid system [36]. Growing evidence highlighted the suitability of BCP for the treatment of chronic inflammation [32], such as the one deriving from osteoarthritis.

Although less documented than BCP, myrcene has also demonstrated its anti-inflammatory properties. In a study conducted by Shamsul et al., this molecule reduced pro-inflammatory cytokines (IL-1β, IL-6, and TNF-α), immunomodulatory factors (interferon gamma (IFNγ), NF-κB and anti-inflammatory markers [interleukin-4 (IL-4), and interleukin-10 (IL-10)] [39]. Interestingly, myrcene is able to act on Transient Receptor Potential Vanilloid 1 (TRPV1), suggesting its potential analgesic action [40]. Similar studies regarding other dietary supplements used to reduce OA pain achieved results in pain control at short follow-ups which were comparable to ours. In particular, a trial that tested the effectiveness of a collagen peptide-based supplement in reducing OA-related lower back pain recorded a 4.1-points reduction according to the Visual Analogical Scale (VAS) in the treatment group after only 3 weeks of intake [16].

Similarly, a recent 8-week randomized double-blind placebo-controlled trial by Wang et al. reached a 2.6-points pain relief according to the Western Ontario and McMaster University (WOMAC) Osteoarthritis index using oral low molecular weight hyaluronic acid in combination with glucosamine and chondroitin on KOA in patients with mild knee pain [41]. Moreover, such evidence is increasingly present in the available literature for supplements apparently less specific for cartilage, but equally valid for the well-known anti-inflammatory power. A systematic review of the nutritional supplement Perna Canaliculus (green-lipped mussel) in the treatment of OA revealed that this molecule could achieve great pain relief at short follow-up according to its ability to counteract joint inflammatory processes [42]. Likewise, cannabidiol demonstrated very encouraging results in counteracting OA-related pain and joint degeneration in both animal models [43] and human studies [44]. Thus, the pain relief we obtained seems to be in line with these results, and was justified by the progressive control of the underlying inflammation.

With regard to the functional outcomes, the ODI and SF-12 scales did not differ appreciably between the two groups, while the KOOS and OKS scales improved significantly in the treatment group. We consider this as the consequence of the fact that ODI and SF-12 are nonspecific for knee evaluation, while KOOS and OKS, as the name suggests (Knee Injury and Osteoarthritis Scale and Oxford Knee Scale), are extremely specific for the pathology we analyzed, which is KOA. Indeed, as is well known, ODI proved to be more suitable for the functional evaluation of spinal disorders [45], while SF-12, which investigates the quality of life, remained less specific and more easily influenced by other factors, especially psychological ones [46]. Furthermore, Table 4 and Table 5 suggest that BMI has a relevant impact on these scales, confirming their nonspecific nature.

We assume that the obtained knee functional improvement is due to the fact that a pain-free joint works better. In fact, one of the first goals in the rehabilitation of knee diseases is precisely to intervene on pain in order to improve joint function and range of motion (ROM) as early as possible [47]. Bahr Taylor et al. showed that massages with essential oils (composed of 55 percent of BCP) on the hands of rheumatoid arthritis patients relieved pain, improved finger strength and significantly increased the angle of maximum flexion compared to subjects treated with coconut oil [48]. Moreover, a study conducted by Topp Robert et al. demonstrated how topical treatment with menthol, which is a terpene as well, improved pain and also knee function in patients with KOA compared to the placebo-treated group [49]. In line with our results, a 2020 in vitro and in vivo study demonstrated the anti-inflammatory action of geranol, an acyclic monoterpene which, when taken per os, protects cartilage and improves joint function [50]. Similarly, as early as 2005, the role of ginger extract in suppressing chemokine induction in human synoviocytes was clear [27], and it was then shown to counteract disability and improve functional capacity in adults with OA [51].

Finally, the dietary supplement containing terpenes appears to have useful properties to improve KOA symptoms. It is likely that the beneficial effects of each nutraceutical component are synergistically amplified by combining them. It is important to further investigate the usefulness of nutraceutical supplements as a complementary treatment for OA because they represent a natural alternative to anti-inflammatory drugs with essentially no side effects that can be self-administered by patients [52,53].

The main limitation of this study is the short duration of follow-up. For this reason, further studies are needed to monitor the durability of the benefits noted in the short term over time. Moreover, the outcome measures are self-reported by patients, but it is a mandatory condition when it is necessary to investigate joint function in the activities of daily living. Finally, there was not a placebo group, but this choice was due to the necessity of guaranteeing a treatment for all patients, since all of them suffered from significant knee pain. Nevertheless, future studies will overcome this limitation, in compliance with the necessary ethical rules.

## 5. Conclusions

In conclusion, the dietary supplement containing terpenes in addition to hemp seed oil is an effective complementary treatment option in patients with KOA for relieving pain and improving joint function. Further studies are needed to prove its efficacy inside multimodal therapies and with longer follow-up periods.

## Figures and Tables

**Figure 1 medicina-59-00191-f001:**
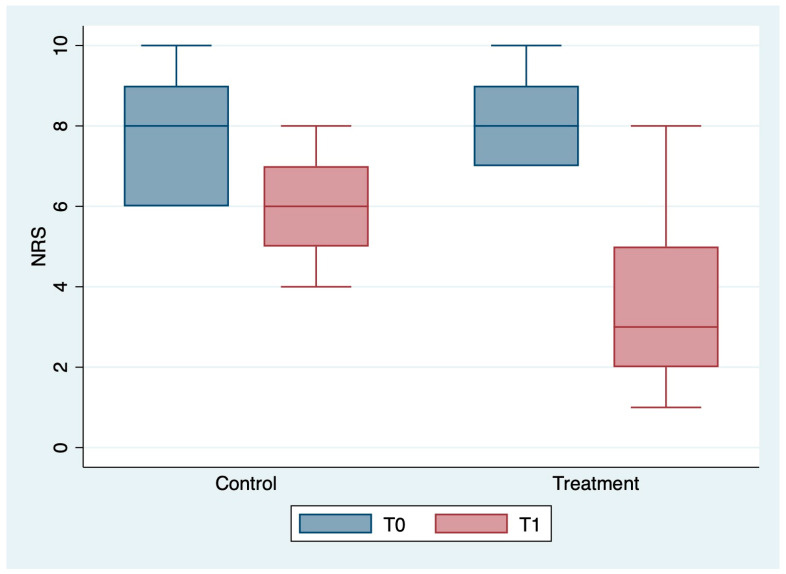
Box plot of NRS values by detection time and group.

**Figure 2 medicina-59-00191-f002:**
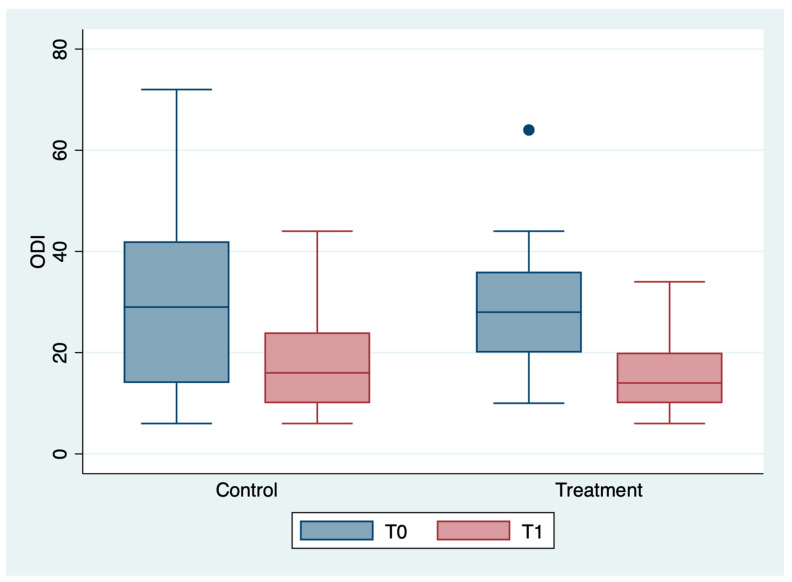
Box plot of ODI values by detection time and group.

**Figure 3 medicina-59-00191-f003:**
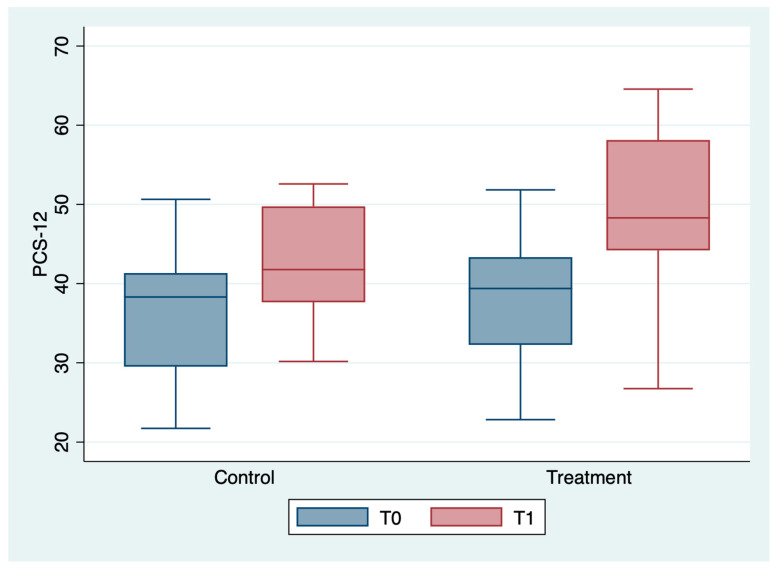
Box plot of PCS-12values by detection time and group.

**Figure 4 medicina-59-00191-f004:**
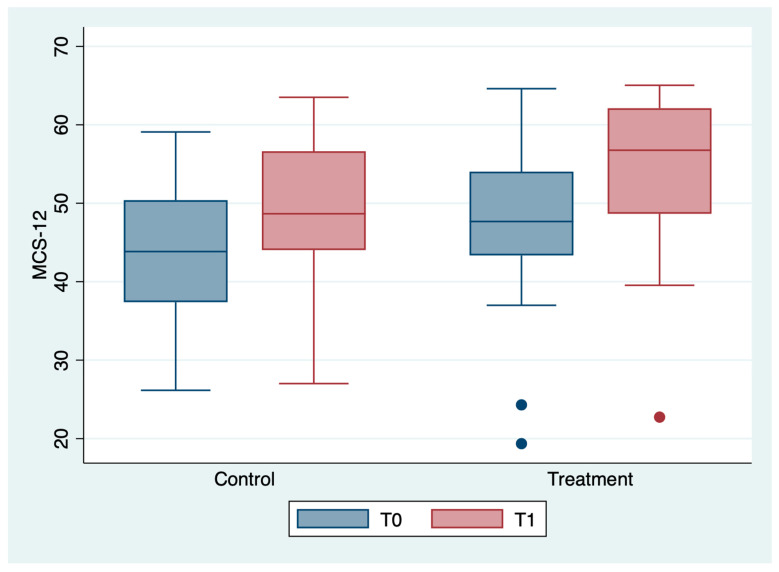
Box plot of MCS-12 values by detection time and group.

**Figure 5 medicina-59-00191-f005:**
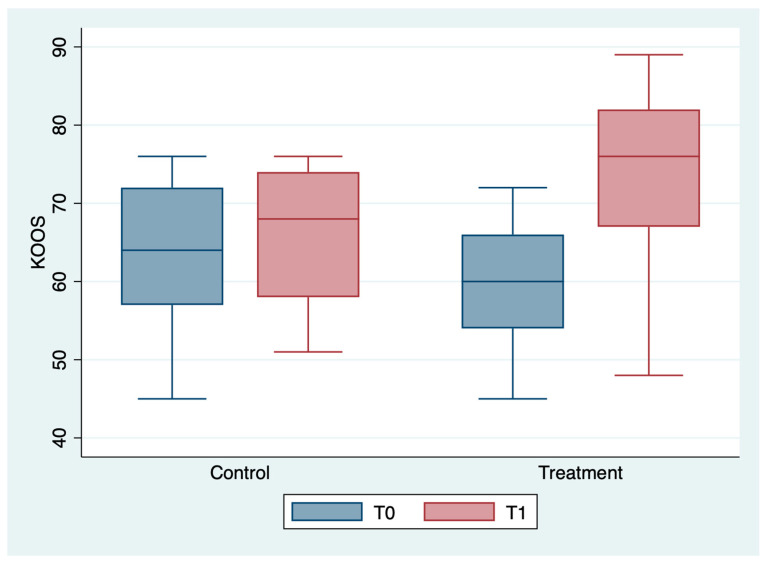
Box plot of KOOS values by detection time and group.

**Figure 6 medicina-59-00191-f006:**
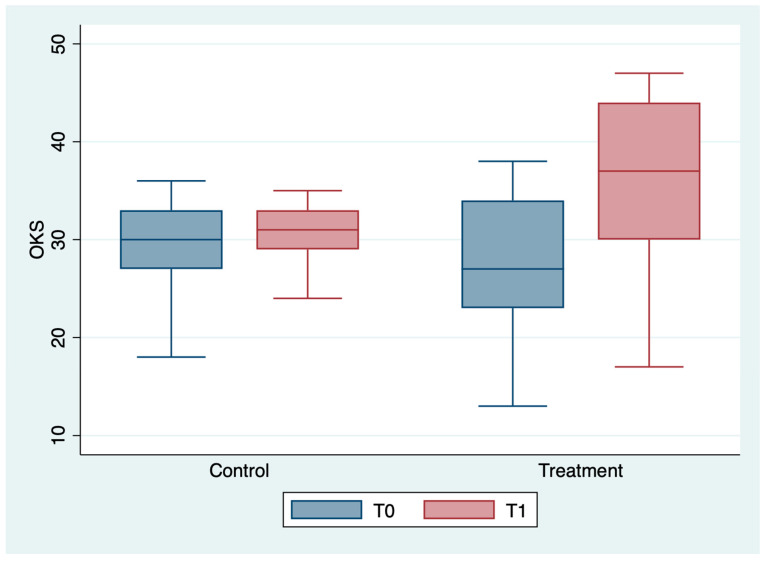
Box plot of OKS values by detection time and group.

**Table 1 medicina-59-00191-t001:** Characteristics of the sample, by group.

Variable	Control (*n* = 19)	Treatment (*n* = 19)	Total (*n* = 38)	*p*-Value
Females; *n* (%)	10 (52.6)	10 (52.6)	20 (52.6)	1.000
Age(years); mean ± SD (range)	59.7 ± 6.6 (47–69)	54.5 ± 4.6 (48–65)	57.1 ± 6.2 (47–69)	0.008
BMI; mean ± SD (range)	27.9 ± 3.8 (20.4–33.4)	29.6 ± 6.3 (20.0–49.3)	28.7 ± 5.2 (20.0–49.3)	0.376

Control = control group; treatment = treatment group; BMI = Body Mass Index; SD = Standard Deviation; *n* = number.

**Table 2 medicina-59-00191-t002:** Mean ± SD and range of outcome variables, by group and detection time.

Variable	Control (*n* = 19)	Treatment (*n* = 19)	Total (*n* = 38)	Group Comparison	Time Comparison	Time and Group Interaction
NRS T0	7.6 ± 1.4 (6–10)	8.3 ± 1.1 (7–10)	7.9 ± 1.3 (6–10)	0.080	<0.0001	<0.0001
NRS T1	5.7 ± 1.2 (4–8)	3.5 ± 2.1 (1–8)	4.6 ± 2.0 (1–8)			
ODI T0	31.3 ± 18.9 (6–72)	29.8 ± 12.8 (10–64)	30.6 ± 15.9 (6–72)	0.687	<0.0001	0.963
ODI T1	17.2 ± 9.3 (6–44)	15.9 ± 7.2 (6–34)	16.6 ± 8.2 (6–44)			
PCS12 T0	36.5 ± 8.4 (21.7–50.6)	37.6 ± 7.7 (22.8–51.8)	37.1 ± 8.0 (21.7–51.8)	0.045	<0.0001	0.066
PCS12 T1	42.6 ± 6.9 (30.2–52.6)	50.2 ± 10.0 (26.7–64.6)	46.4 ± 9.3 (26.7–64.6)			
MCS12 T0	43.3 ± 10.1 (26.2–59.1)	46.5 ± 10.7 (19.4–64.6)	44.9 ± 10.4 (19.4–64.6)	0.138	<0.0001	0.190
MCS12 T1	47.9 ± 10.7 (27.0–63.5)	54.3 ± 10.3 (22.7–65.0)	51.2 ± 10.8 (22.7–65.0)			
KOOS T0	62.9 ± 9.2 (45–76)	59.8 ± 7.1 (45–72)	61.4 ± 8.2 (45–76)	0.403	<0.0001	<0.001
KOOS T1	66.4 ± 8.6 (51–76)	74.0 ± 11.0 (48–89)	70.2 ± 10.5 (58–89)			
OKS T0	29.4 ± 4.8 (18–36)	27.4 ± 6.3 (13–38)	28.4 ± 5.6 (13–38)	0.278	<0.0001	<0.0001
OKS T1	30.6 ± 3.1 (24–35)	36.6 ± 8.2 (17–47)	33.6 ± 6.8 (17–47)			

Control = control group; treatment = treatment group; *n* = number; NRS = Numerical Rating Scale; ODI = Oswestry Disability Index; PCS12 = SF12 Physical Component Dimension; MCS12 = SF12 Mental Component Dimension); KOOS = Knee Injury and Osteoarthritis Outcome Score; OKS = Oxford Knee Score.

**Table 3 medicina-59-00191-t003:** Analysis of the determinants of the difference between NRS T1 and NRS T0 in a multivariate linear regression model.

Determinants	Coef.	95%CI	*p*-Value
Group (treatment vs. control)	−2.8	−4.0–1.6	<0.0001
Sex (male vs. female)	0.04	−1.08–1.17	0.938
Age (years)	0.04	−0.06–0.14	0.404
BMI	0.04	−0.07–0.15	0.486

BMI = Body Mass Index; Coef = coefficient; CI = Confidence Interval.

**Table 4 medicina-59-00191-t004:** Analysis of the determinants of the difference between ODI T1 and ODI T0 in a multivariate linear regression model.

Determinants	Coef.	95%CI	*p*-Value
Group (treatment vs. control)	0.89	−9.0–10.8	0.856
Sex (male vs. female)	−4.70	−13.8–4.4	0.303
Age (years)	−0.19	−0.98–0.61	0.640
BMI	−0.97	−1.88–−0.07	0.036

BMI = Body Mass Index; Coef = coefficient; CI = Confidence Interval.

**Table 5 medicina-59-00191-t005:** Analysis of the determinants of the difference between PCS-12 T1 and PCS-12 T0 in a multivariate linear regression model.

Determinants	Coef.	95%CI	*p*-Value
Group (treatment vs. control)	8.5	0.6–16.3	0.036
Sex (male vs. female)	0.6	−6.6–7.8	0.864
Age (years)	0.2	−0.4–0.9	0.409
BMI	−0.4	−1.1–0.3	0.036

BMI = Body Mass Index; Coef = coefficient; CI = Confidence Interval.

**Table 6 medicina-59-00191-t006:** Analysis of the determinants of the difference between MCS-12 T1 and MCS-12 T0 in a multivariate linear regression model.

Determinants	Coef.	95%CI	*p*-Value
Group (treatment vs. control)	3.7	−2.3–9.6	0.218
Sex (male vs. female)	−0.03	−5.50–5.44	0.992
Age (years)	0.10	−0.38–0.58	0.674
BMI	0.08	−0.46–0.62	0.769

BMI = Body Mass Index; Coef = coefficient; CI = Confidence Interval.

**Table 7 medicina-59-00191-t007:** Analysis of the determinants of the difference between KOOS T1 and KOOS T0 in a multivariate linear regression model.

Determinants	Coef.	95%CI	*p*-Value
Group (treatment vs. control)	11.9	5.8–18.0	<0.0001
Sex (male vs. female)	0.7	−4.9–6.3	0.792
Age (years)	0.2	−0.3–0.6	0.515
BMI	−0.2	−0.8–0.3	0.402

BMI = Body Mass Index; Coef = coefficient; CI = Confidence Interval.

**Table 8 medicina-59-00191-t008:** Analysis of the determinants of the difference between OKS T1 and OKS T0 in a multivariate linear regression model.

Determinants	Coef.	95%CI	*p*-Value
Group (treatment vs. control)	8.5	5.4–11.7	<0.0001
Sex (male vs. female)	0.4	−2.5–3.3	0.796
Age (years)	0.1	−0.2–0.3	0.511
BMI	−0.1	−0.4–0.2	0.537

BMI = Body Mass Index; Coef = coefficient; CI = Confidence Interval.

## Data Availability

The datasets used and analyzed during the current study will be made available upon reasonable request to the corresponding author, G.F.

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
