# Peer review of "Hemp Seed Oil in Association with β-Caryophyllene, Myrcene and Ginger Extract as a Nutraceutical Integration in Knee Osteoarthritis: A Double-Blind Prospective Case-Control Study"

_medicina, 2023, doi:10.3390/medicina59020191_

Round 1

Reviewer 1 Report

The article: “Hemp Seed Oil in Association with Β-Caryophyllene, Myrcene 2 and Ginger Extract as a Nutraceutical Integration in Knee Osteoarthritis: A Double-Blind Prospective Case-Control Study”
is nicely written and organized, and even if the results are not strong they could be an inspiration for further insights.
So I have a few comments and suggestions for the authors:

INTRO:
lines 135-138: “Patients belonging to treatment group underwent a 45 days treatment with a dietary supplement based on hemp seed oil and β-caryophyllene, myrcene and ginger extract titrated in gingerols”
could you please specify clearly the composition of your products (how much β-caryophyllene, myrcene, and ginger extract are present), is it known?
moreover, I would suggest mentioning for the β-caryophyllene these two studies:
i)
In Vitro Effects of Low Doses of β-Caryophyllene, Ascorbic Acid and D-Glucosamine on Human Chondrocyte Viability and Inflammation; Mattiuzzo et al.; Pharmaceuticals 2021, 14, 286
ii) β-Caryophyllene Mitigates Collagen Antibody Induced Arthritis (CAIA) in Mice Through a Cross-Talk between CB2 and PPAR-γ Receptors. Irrera, N. et al; Biomolecules 2019, 9, 326

could you please add a couple of sentences for the ginger extract mentioned only after in the methods section? Why did you choose to add it?
Is your [47] reference worth inserting here?

RESULTS AND STATISTICS:

Table 2 and Figure 3:
PCS12: since the two groups are significative different already in the beginning, even if the amelioration is astonishing
I would strongly recommend you to expand your explanation on this variable. Thank you

Table 2 and Figure
MCS12: the two groups appear really different already from the beginning looking at the graph and without any difference from the table, I would strongly recommend you to expand your explanation on this variable. Thank you

Numbers on graphs and numbers on the tables look like an incongruence: is it possible to show a Spaghetti-box plot for these data?  (a box plot with
Lines connecting the results obtained from the same subject)

DISCUSSION

Lines 249-252:
I would suggest a recent in vitro study supporting your work that gives new insights into the activity of BCP, and its antioxidant, anti-inflammatory, and chondroprotective effects that will support your work:
In Vitro Effects of Low Doses of β-Caryophyllene, Ascorbic Acid and D-Glucosamine on Human Chondrocyte Viability and Inflammation; Mattiuzzo et al.; Pharmaceuticals 2021, 14, 286.

Line 264: “Reduction of acute and chronic pain is achieved by BCP” since you recorded that, did you observe a difference in paracetamol intake between the two groups only hemp vs hemp + BCP?

could the BCP, myrcene, and ginger have an effect by themselves or is it synergic with hemp?

An interesting evaluation for a future study, in place or in addition to the “placebo” group could be
to study BCP, myrcene, and ginger diluted in a different oil.

Author Response

Dear Reviewer,

Thank You for Your precious comments which certainly will improve the quality of our paper.

INTRO:

Q1: lines 135-138: “Patients belonging to treatment group underwent a 45 days treatment with a dietary supplement based on hemp seed oil and β-caryophyllene, myrcene and ginger extract titrated in gingerols”

could you please specify clearly the composition of your products (how much β-caryophyllene, myrcene, and ginger extract are present), is it known?

A1: thank You for pointing out this concern, we integrated the text as You required.

Q2: moreover, I would suggest mentioning for the β-caryophyllene these two studies:

  1. i) In Vitro Effects of Low Doses of β-Caryophyllene, Ascorbic Acid and D-Glucosamine on Human Chondrocyte Viability and Inflammation; Mattiuzzo et al.; Pharmaceuticals 2021, 14, 286
  2. ii) β-Caryophyllene Mitigates Collagen Antibody Induced Arthritis (CAIA) in Mice Through a Cross-Talk between CB2 and PPAR-γ Irrera, N. et al; Biomolecules 2019, 9, 326

A2: Thank You for the suggestion, we integrated our references with the studies You indicated.

Q3: could you please add a couple of sentences for the ginger extract mentioned only after in the methods section? Why did you choose to add it?  Is your [47] reference worth inserting here?

Thank You for this question. Ginger extract was included for the anti-inflammatory properties of gingerols and shagaols which takes place through a selective inhibition of COX-2, factor NF-Kb, iNOS, IL-1, IL-6 and TNF-α. Thus it enhances the effect of the other components of the nutraceutical we tested. So, we explained it in the introduction, placing the reference [47] in a more suitable position in the text.

RESULTS AND STATISTICS:

Q4: Table 2 and Figure 3:

PCS12: since the two groups are significative different already in the beginning, even if the amelioration is astonishing. I would strongly recommend you to expand your explanation on this variable. Thank you

Table 2 and Figure

MCS12: the two groups appear really different already from the beginning looking at the graph and without any difference from the table, I would strongly recommend you to expand your explanation on this variable. Thank you

A4: thank You for this question, we provided to better explain these data in the results, especially with regard to MCS12, that is easily influenced by factors other than knee pain. We also explained it later in the discussion.

Q5: Numbers on graphs and numbers on the tables look like an incongruence: is it possible to show a Spaghetti-box plot for these data?  (a box plot with Lines connecting the results obtained from the same subject)

A5: Thank You for this comment. We completely agree with your consideration, but unfortunately STATA software does not allow to reproduce Spaghetti-box plots. Nevertheless, we replaced all the figures with box plots in order to make these data clearer to the readers.

DISCUSSION

Q5: Lines 249-252:

I would suggest a recent in vitro study supporting your work that gives new insights into the activity of BCP, and its antioxidant, anti-inflammatory, and chondroprotective effects that will support your work:

In Vitro Effects of Low Doses of β-Caryophyllene, Ascorbic Acid and D-Glucosamine on Human Chondrocyte Viability and Inflammation; Mattiuzzo et al.; Pharmaceuticals 2021, 14, 286.

A5: Thank You for your suggestion, we reported this precious reference in the discussion, as You required.

Q6: Line 264: “Reduction of acute and chronic pain is achieved by BCP” since you recorded that, did you observe a difference in paracetamol intake between the two groups only hemp vs hemp + BCP?

A6: Thank You for Your question. As we reported in the manuscript, there were no significant differences in the paracetamol intake between the groups, and the intake was sporadic and not significant.

Q7: could the BCP, myrcene, and ginger have an effect by themselves or is it synergic with hemp?

A7: Although each component has a specific beneficial action, we believe that the effects are positively amplified by combining them. We briefly integrated the discussion whit this statement.

Q8: An interesting evaluation for a future study, in place or in addition to the “placebo” group could be to study BCP, myrcene, and ginger diluted in a different oil.

A8: Thank You for this suggestion. We agree with this consideration, and we will try to overcome this limitation in our future related studies.

Thank You again for the time You dedicated to our paper

Best regards

Reviewer 2 Report

Commnets

1. Lines 135-143: The amounts of constituents administered per day should be clearly described in control group and treatment group.

2. Lines 144-245: The reviewer is not sure whether one soft gel id one capsule, and the supplement was administered before, after or at the meal.

3. In the Figures 1-6, the statistic differences should be described between T0 and T1, between control and treatment groups at T0, and between control and treatment groups at T1. That is easily understood by the readers.

Author Response

Dear Reviewer,

Thank you for your kind comments and for precious suggestions, which certainly will help us to improve the quality of our paper.

Q1: 1. Lines 135-143: The amounts of constituents administered per day should be clearly described in control group and treatment group.

A1: Thank You for pointing out this concern. We integrated the text with all the data You required.

Q2. Lines 144-245: The reviewer is not sure whether one soft gel id one capsule, and the supplement was administered before, after or at the meal.

A2: Thank You for pointing out this doubt. We used softgel capsules which were taken during meals. We integrated the text with these data.

Q3: In the Figures 1-6, the statistic differences should be described between T0 and T1, between control and treatment groups at T0, and between control and treatment groups at T1. That is easily understood by the readers.

A3: Thank You for this suggestion, we provided to explain clearer the results represented in the figures.

Thank You again for the time you dedicated to our paper.

Best regards